# SummIt: Iterative Text Summarization via ChatGPT

**Haopeng Zhang      Xiao Liu      Jiawei Zhang**

IFM Lab, Department of Computer Science, University of California, Davis, CA, USA

`haopeng,xiao,jiawei@ifmlab.org`

## Abstract

Text summarization systems have made significant progress in recent years, but typically generate summaries in one single step. However, the one-shot summarization setting is sometimes inadequate, as the generated summary may contain hallucinations or overlook essential details related to the reader's interests. This paper addresses this limitation by proposing SummIt, an iterative text summarization framework based on large language models like ChatGPT. Our framework enables the model to refine the generated summary iteratively through self-evaluation and feedback, resembling humans' iterative process when drafting and revising summaries. Furthermore, we explore the potential benefits of integrating knowledge and topic extractors into the framework to enhance summary faithfulness and controllability. We automatically evaluate the performance of our framework on three benchmark summarization datasets. We also conduct a human evaluation to validate the effectiveness of the iterative refinements and identify a potential issue of overcorrection. [1]

## 1 Introduction

Document summarization aims to compress text material while retaining its most salient information. With the increasing amount of publicly available text data, the significance of automated summarization techniques has amplified. Recent advancements in summarization systems, leveraging neural networks and pre-trained language models, have demonstrated notable progress (Cheng and Lapata, 2016; Nallapati et al., 2016; Liu and Lapata, 2019; Lewis et al., 2019; Zhang et al., 2020). The above summarization systems are all built end-to-end and follow a one-shot paradigm that generates summaries in a single step. On the contrary, humans often write text through an evolutionary pro-

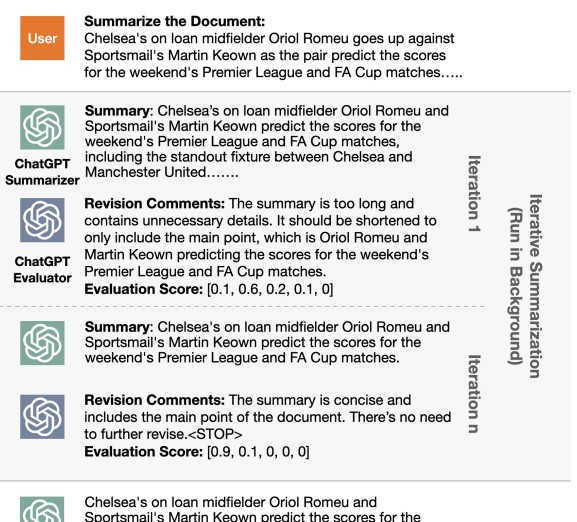

Figure 1: An illustration of the iterative summarization process. The summarizer continuously refines the summary according to self-feedback from the evaluator at each iteration.

cess characterized by multiple iterations of drafting and editing (Faltings et al., 2020).

These end-to-end summarization systems encounter multiple challenges. Firstly, they frequently suffer from the issue of hallucination, resulting in the generation of ungrammatical or factually incorrect content (Kryscinski et al., 2020; Zhang et al., 2022b). Secondly, these systems are often optimized using imperfect reference summaries, and widely adopted evaluation metrics like ROUGE may not accurately assess summary quality. Thirdly, most of these systems lack controllability, as they only produce a single generic summary conditionally on the input document. In practice, instead of a single condensed version of the entire document, generating summaries that cater to specific aspects or queries would be more beneficial to meet the diverse requirements of users.

The emergence of advanced instruction-tuned

---

[1]We will release our codebase at `https://github.com/hpzhang94/summ_it`

large language models (LLMs), such as ChatGPT [2], has presented exciting possibilities for summarization systems by exhibiting strong zero-shot performance in various downstream tasks. A recent study by (Goyal et al., 2022) compared GPT-3 with traditional fine-tuning methods and found that despite lower ROUGE scores, human annotators preferred the GPT-3 generated summaries. Another comprehensive analysis by (Zhang et al., 2023d) focused on large language models for news summarization and revealed that the quality of generated summaries is already on par with those created by humans. Furthermore, Liu et al. (2023) demonstrated the utilization of LLMs like GPT-4 as an effective natural language generation evaluator, showing a higher correlation with humans in the summarization task compared to previous reference-based methods.

The advent of LLMs also introduces new opportunities for summarization beyond the traditional one-shot generation setting. In this paper, we introduce **SummIt**, a framework that leverages large language models for **it**erative text **summ**arization. Instead of generating summaries in a single step, our framework enables the model to iteratively refine the generated summary through self-evaluation and feedback, resembling the human process of drafting and revising summaries. According to our experiments, the rationale generation and summary refinement in SummIt can be guided effectively with in-context learning, eliminating the need for supervised training or reinforcement learning processes. Additionally, we explore the potential benefits of incorporating **knowledge** and **topic** extractors to enhance summary faithfulness and controllability. We instantiate SummIt with ChatGPT as the backbone, and the automatic evaluation results on three benchmark datasets demonstrate the effectiveness of SummIt in improving summary quality, faithfulness, and controllability within only a few iterations. Furthermore, we conduct a human evaluation to validate the iterative refinements quality and identify a potential over-correction issue.

We summarize the contributions of this paper as follows:

- We propose SummIt, a novel framework for iterative text summarization. SummIt enables the iterative refinement of generated summaries by incorporating self-evaluation and

feedback mechanisms. In addition, we propose to incorporate knowledge and topic extractors to further improve the faithfulness and controllability of SummIt.

- We conduct experiments on three summarization benchmark datasets, and empirical results from automatic evaluation demonstrate the effectiveness of our proposed framework in summary refinement.

- A human evaluation is conducted to investigate the impact of self-evaluation-guided summary refinement. The results revealed a potential issue of **over-correction**: while the large language model effectively refines the summary based on the feedback rationale, it exhibits a bias toward its own evaluation criteria, rather than aligning closely with human judgment.

## 2 Related Work

### 2.1 Text Summarization

Recent years have witnessed significant advancements in text summarization systems with the development of deep neural networks and pre-trained language models. Automatic summarization methods can be broadly categorized into extractive and abstractive approaches. Extractive summarization involves the direct extraction of sentences from the source text to form summaries (Xu et al., 2019; Liu and Lapata, 2019; Zhong et al., 2020; Zhang et al., 2022a, 2023b), while abstractive approaches conditionally generate summaries using a sequence-to-sequence (seq2seq) framework (Lewis et al., 2019; Zhang et al., 2020).

Existing approaches mentioned above generate summaries in a one-shot manner, and their outputs may not always align with user expectations and may contain hallucinated content (Kryscinski et al., 2020). To address the limitation, Liu et al. (2022) proposes to automatically correct factual inconsistencies in generated summaries with generated human feedback. In contrast, our SummIt framework enables iterative summary refinement with self-evaluation and feedback, eliminating the need for costly human annotations. Additionally, we propose the integration of knowledge and topic extractors to further enhance summary faithfulness and controllability.

[2]https://chat.openai.com/chat

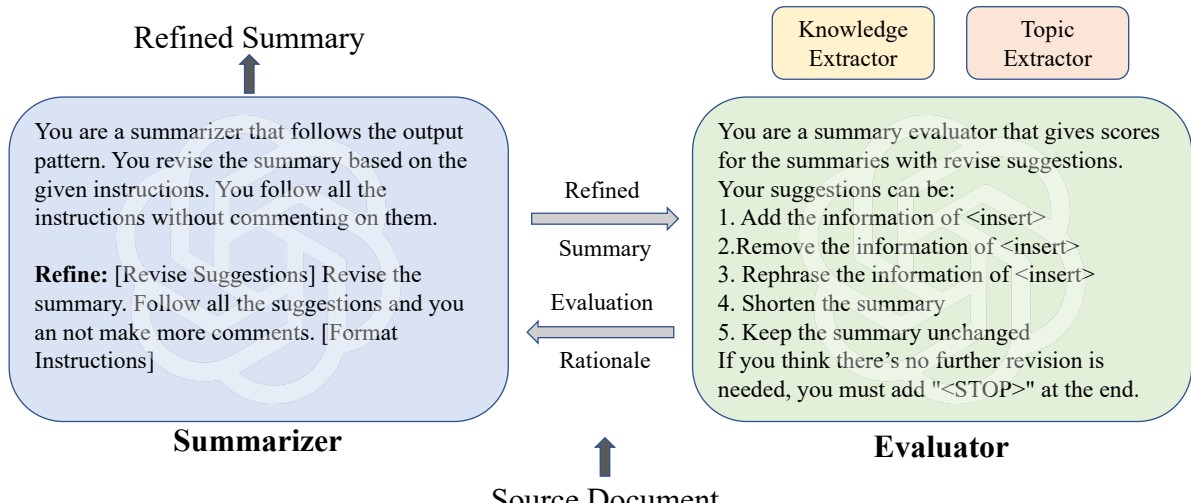

Figure 2: The overall framework of our proposed iterative text summarization system. The evaluator generates an evaluation rationale based on the current summary, and the summarizer then refines the summary accordingly. The knowledge and topic extractors retrieve information from the source document to guide the process.

## 2.2 Summarization with Large Language Models

Recent years have seen a surge in training large-scale language models (LLM) on large amounts of text, such as GPT (Radford et al., 2019; Brown et al., 2020). Several studies have explored the application of LLMs in the context of text summarization. For instance, Goyal et al. (2022) compared the performance of GPT-3-generated summaries with traditional fine-tuning methods, finding that although the former achieved slightly lower ROUGE scores, human evaluators expressed a preference for them. Similarly, Zhang et al. (2023d) reported that LLM-generated summaries were on par with human-written summaries in the news domain. Zhang et al. (2023c) benchmarked the performance of ChatGPT on extractive summarization and proposes to improve summary faithfulness with an extract-then-generate pipeline. On the other hand, prior works have also leveraged LLMs for summarization evaluation (Liu et al., 2023; Fu et al., 2023; Luo et al., 2023), demonstrating that LLM-based metrics outperform all previous evaluation metrics like ROUGE (Lin, 2004) and BertScore (Zhang et al., 2019) by a significant margin in terms of correlation with human evaluations.

## 2.3 Text Editing

Our work is also closely related to the task of text editing. Traditional editing models are trained to solve specific tasks, such as information updating (Iso et al., 2020), Wikipedia edit (Reid and Neu-

big, 2022), and grammar error correction (Awasthi et al., 2019). Recent works also formulate text editing as an interactive task, such as command-based editing systems (Faltings et al., 2020), and interactive editing systems (Schick et al., 2022). Zhang et al. (2023a) also proposed a benchmark for fine-grained instruction-based editing.

Recently, Welleck et al. (2022) introduced a self-corrective learning framework that incorporates a corrector into the language model to facilitate self-correction during sequence generation. Akyürek et al. (2023) propose a reinforcement learning-based approach to generate natural language feedback for correcting generation errors. Concurrent work Madaan et al. (2023) presents a similar generation pipeline that enhances initial outputs through iterative feedback using a single LLM for short text generation tasks. In contrast, our SummIt framework differs from these approaches as it specifically focuses on the conditional generation task of summarization, with an emphasis on improving summary faithfulness and controllability. Additionally, we empirically observe that separating the summarizer and evaluator into different LLMs, each employing different in-context guidance leads to improved performance in our framework.

## 3 Methods

## 3.1 Iterative Summarization

The overall architecture of our iterative text summarization system SummIt is shown in Figure 2.

The system consists of two major components, a summarizer that generates and refines the summary, and an evaluator that generates feedback rationale.

**Summarizer:** The summarizer is in charge of generating the initial summary and revising a summary conditioned on the given explanations and source document. We instantiate the summarizer with an instruction-tuned language model $S$.

Formally, given the input source document $\mathbf{x}$, the initial summary $\mathbf{y}^0$ generation process can be represented as:

$$p_S(\mathbf{y}^0 \mid \mathbf{x}) = \prod_{t=1}^{m} p_S\left(y_t^0 \mid \mathbf{y}_{<t}^0, \mathbf{x}\right), \quad (1)$$

, where $\mathbf{y}_{<t}^0$ denotes the generated tokens, $y_t^0$ refers to the $t$-th summary token, and $m$ denotes the summary length.

After obtaining the $i$-step self-evaluation feedback $\mathbf{e}^i$ from the evaluator $E$, the summarizer will refine the summary accordingly and then generates refined summary $\mathbf{y}^{(i+1)}$ as: $p_S(\mathbf{y}^{(i+1)} \mid \mathbf{x}, \mathbf{e}^i)$.

**Evaluator:** The evaluator is another instance of language model $E$ that generates summary quality evaluation and corresponding explanations $\mathbf{e}^i$ for the $i$-th iteration as: $p_E(\mathbf{e}^i \mid \mathbf{x}, \mathbf{y}^i)$.

**Stopping Criteria:** The evaluator gives a quality assessment of the generated summary and then outputs the rationale for the evaluation as feedback. The summarizer receives model evaluation and feedback from the evaluator, subsequently refining the summary based on this input.

This iterative process can be repeated until **1)** the evaluator determines that no further refinement is required or **2)** fulfills rule-based stopping criteria, such as reaching a maximum iteration number.

## 3.2 In-context Learning

Since the summarizer and evaluator in SummIt are not fine-tuned with supervised data or trained reinforcement learning rewards, it would be beneficial to guide the explanation and summary generation process with the desired format or template. Recent studies have shown that large language models have strong few-shot performance on various downstream tasks, known as in-context learning (ICL) (Brown et al., 2020).

The standard ICL prompts a language model, $M$, with a set of exemplar source-summary pairs,

| Dataset | #Test | Doc #words | Sum #words | #Sum |
|---------|-------|------------|------------|------|
| XSum | 11,334 | 430.2 | 23.3 | 1 |
| CNN/DM | 11,489 | 766.1 | 58.2 | 1 |
| NEWTS | 600 | 738.5 | 70.1 | 2 |

Table 1: Detailed statistics of the experimental datasets. Doc # words and Sum # words refer to the average word number in the source document and summary.

$\mathbf{C} = \{(x_1, y_1)...(x_m, y_m)\}$, and generates summary $\mathbf{y}$ by concatenating the exemplar source-summary pairs and input document as prompt: $p_M(\mathbf{y} \mid \mathbf{x}, \mathbf{C})$.

We also use in-context learning to guide our iterative summarization system, where we use "document-reference summary" pairs as the context for the summarizer $S$, and use "document-reference summary-human written explanation" triplets as the context for the evaluator $E$. We empirically find that in-context learning could improve the efficacy of our system.

## 3.3 Summary Faithfulness and Controllability

In practical applications, the faithfulness of the generated summary holds significant importance, alongside the overall quality of the summary (Kryscinski et al., 2020). Previous research has demonstrated the effectiveness of leveraging knowledge extraction from source documents to enhance the faithfulness of generated summaries (Huang et al., 2020; Zhu et al., 2020). Building upon these insights, we propose integrating a knowledge extractor into our iterative summarization system.

**Knowledge Extractor:** In particular, we utilize OpenIE [3], which extracts knowledge $\mathbf{k}$ in the form of triplets from the source document. During each iteration, the summarizer ($S$) is guided to refine the summary in accordance with the extracted knowledge, represented as: $p_S(\mathbf{y}^{(i+1)} \mid \mathbf{x}, \mathbf{e}^i, \mathbf{k})$. Moreover, the evaluator ($E$) can be directed to factor in faithfulness when delivering feedback, denoted as $p_E(\mathbf{e}^i \mid \mathbf{x}, \mathbf{y}^i, \mathbf{k})$, as LLMs have shown to be efficient faithfulness evaluators (Luo et al., 2023).

Furthermore, real-world applications often require the generation of summaries tailored to specific aspects or queries, rather than a single generic summary of the entire document. Our iterative summarization framework offers enhanced controllability for aspect-based summarization tasks.

---

[3] https://stanfordnlp.github.io/CoreNLP/openie.html

**Topic Extractor:** Given an aspect-oriented query $q$, we prompt both summarizer $S$ and evaluator $E$ to initially extract relevant snippets, each containing less than 5 words, from the source document **x**. Following the extraction, these components then proceed to either generate or assess the summary by taking into account the extracted snippets. The iterative nature of our framework further facilitates the controllable summary generation, allowing for the easy transformation of generic summaries into topic-focused summaries based on the user's preferences.

## 3.4 Prompt Format

We utilize both system prompts and user prompts following the OpenAI API in our system implementations. The full prompts used in the experiments can be found in Table 8 and Table 9. Notably, we empirically find that pre-defining the possible edit operations for the evaluator improves the system performance significantly since it avoids free-form edits to the summary by the large language model. Thus, we adopt the five types of text editing operations commonly used in text editing systems (Reid and Neubig, 2022; Faltings et al., 2020). We specifically require the evaluator to generate feedback based on the source document and summary at this iteration with the following five types of possible refinement operations:

- **Add**: Add the information of <insert>

- **Remove**: Remove the information of <insert> from the summary

- **Rephrase**: Rephrase the information of <insert> in the summary

- **Simplify**: Shorten the summary

- **Keep**: Keep the summary unchanged

## 4 Experiments

In this section, we validate our SummIt framework on three benchmark summarization datasets. We employ both automatic metrics and human assessment to evaluate the quality 4.2, faithfulness 4.3, and controllability 4.4 of the generated summaries.

## 4.1 Experiment Settings

**Datasets:** We conduct experiments on the following three publicly available benchmark datasets, as presented in Table 1, ensuring they are consistent with previous fine-tuning approaches: *1)* *CNN/DailyMail* (Hermann et al., 2015) is the most widely-adopted summarization dataset that contains news articles and corresponding human-written news highlights as summaries. We use the non-anonymized version in all experiments. *2)* *XSum* (Narayan et al., 2018) is a one-sentence news summarization dataset with all summaries professionally written by the original authors of the BBC news. *3) NEWTS* (Bahrainian et al., 2022) is an aspect-focused summarization dataset derived from the CNN/DM dataset and contains two summaries focusing on different topics for the same news.

**Evaluation metrics:** For summary *quality*, we use ROUGE scores (Lin, 2004) and G-Eval (Liu et al., 2023) as the automatic metrics. We report ROUGE-1, ROUGE-2, and ROUGE-L scores, which respectively measure the overlap of unigrams, bigrams, and the longest common sequence between the generated summary and the reference summary. G-Eval is an LLM-based matrix with a scale ranging from 1 to 5. G-Eval uses LLM with chain-of-thoughts (CoT) and a form-filling paradigm to assess the quality of NLG outputs. It shows the highest correlation with humans compared to other summarization quality metrics.

For summary *faithfulness*, we use FactCC (Kryscinski et al., 2020) and DAE (Defining arc entailment) (Goyal and Durrett, 2020) as our evaluation metrics. FactCC is a weakly supervised BERT-based model metric that verifies factual consistency through rule-based transformations applied to source document sentences. It shows a high correlation in assessing summary faithfulness with human judgments. DAE decomposes entailment at the level of dependency arcs, examining the semantic relationships within the generated output and input. Rather than focusing on aggregate decisions, DAE measures the semantic relationship manifested by individual dependency arcs in the generated output supported by the input.

For the *controllability* of query-focused summarization, we use BM25 (Robertson et al., 2009) and DPR(Karpukhin et al., 2020) to measure the similarity between the query and the summary with both sparse and dense evaluations. BM25 is a probabilistic retrieval function that ranks documents based on query term frequency. DPR leverages dense vector representations for scalable retrieval, embedding both questions and passages into fixed-length vector spaces for nuanced similarity calculations.

| Model | CNN/DM | | | | XSum | | | |
|---|---|---|---|---|---|---|---|---|
| | **R1** | **R2** | **RL** | **G-Eval** | **R1** | **R2** | **RL** | **G-Eval** |
| *Zero-shot setting* | | | | | | | | |
| PEGASUS$_{ZS}$ | 32.90 | 13.28 | 29.38 | 3.23 | 19.27 | 3.00 | 12.72 | 3.52 |
| BART$_{ZS}$ | 32.83 | 13.30 | 29.64 | 3.42 | 19.26 | 3.30 | 14.67 | 3.49 |
| T5$_{ZS}$ | **39.68** | **17.24** | 26.28 | 3.47 | 19.66 | 2.91 | 15.31 | 3.55 |
| ChatGPT | 39.44 | 16.14 | **29.83** | 3.46 | 21.61 | **5.98** | 17.60 | 3.47 |
| SummIt (ours) | 36.50 | 13.49 | 26.76 | **4.33** | **21.92** | 5.93 | **17.62** | **4.24** |
| *Few-shot setting* | | | | | | | | |
| ChatGPT | **40.00** | **16.39** | **30.02** | 3.57 | **23.96** | **7.36** | **19.36** | 3.57 |
| SummIt (ours) | 37.29 | 13.60 | 26.87 | **4.35** | 22.04 | 6.20 | 17.46 | **4.32** |

Table 2: This table presents results from experiments conducted on the CNN/DM and XSum datasets under both zero-shot and few-shot settings. A random sample of $1,000$ data points was taken from each dataset for evaluation. G-Eval represents the score evaluated by the ChatGPT evaluator in our framework.

| Model | Coherence | Fluency | Relevance | Consistency | Conciseness | Overall | Human Pref |
|---|---|---|---|---|---|---|---|
| *CNN/DM* | | | | | | | |
| BART | 3.92 | 4.16 | 4.00 | 3.12 | 3.64 | 3.24 | 0.04 |
| T5 | 3.72 | 4.24 | **4.32** | 3.52 | 3.84 | 3.68 | 0.10 |
| PEGASUS | 3.20 | 3.53 | 3.33 | 2.87 | 1.85 | 1.63 | 0.00 |
| ChatGPT | 4.20 | 4.36 | 4.28 | 4.01 | **3.92** | 4.01 | 0.34 |
| SummIt | **4.24** | **4.50** | 4.29 | **4.12** | 3.84 | **4.09** | **0.52** |
| *XSum* | | | | | | | |
| BART | 3.97 | 4.30 | 4.13 | 3.30 | **3.93** | 3.84 | 0.30 |
| T5 | 3.84 | 4.32 | 4.02 | 3.63 | 3.84 | 3.25 | 0.08 |
| PEGASUS | 3.13 | 4.10 | 3.52 | 2.87 | 2.03 | 2.41 | 0.00 |
| ChatGPT | 4.03 | **4.40** | **4.30** | 3.93 | 3.87 | 3.92 | 0.24 |
| SummIt | **4.04** | 4.35 | 4.28 | **4.05** | 3.72 | **3.96** | **0.38** |

Table 3: Human study results on generic summary quality. The first five columns include Likert scale ratings and the last column is the human preference results.

In line with previous research findings that have emphasized the inclination of human annotators towards summaries generated by LLM models, even in the presence of comparatively lower ROUGE scores (Goyal et al., 2022), we further validate the effectiveness of SummIt through a dedicated human study. Specifically, we use **1)** five-point Likert scale ratings (Likert, 1932) covering summary coherence, fluency, relevance, consistency, conciseness, and overall evaluation, and **2)** human preference test, where annotators are shown summaries of the same source document from all five summarization systems and then asked to select their most preferred summary or summaries.

We evaluated the performance using 1000 ran-

dom samples from CNN/DM and XSum test sets, with seed 101, and the full NEWTS test set. Our prompts were refined with a 50-example development set. The detailed experimental setup is provided in Appendix A.

## 4.2 Generic Summary Quality Evaluation

The automatic evaluation results for generic summarization quality are shown in Table 2. We use previous pre-trained language models, including PEGASUS (Zhang et al., 2020), BART (Lewis et al., 2019), and T5 (Raffel et al., 2020) as baseline models. We compare our framework SummIt with these baseline models under a zero-shot setting for a fair comparison.

It is observed that SummIt has inferior ROUGE scores compared to fine-tuning approaches on CNN/DM, while exhibiting significantly higher LLM-based evaluation metric G-Eval. On the other hand, it outperforms all baseline methods on the XSum dataset. Compared to the output of ChatGPT, the summaries of ChatGPT after our iterative refinement see a consistent improvement in the G-Eval score. The results are consistent with the previous conclusions in (Zhang et al., 2023d), where large language model summary outputs receive lower ROUGE scores due to the low quality of reference summaries.

In addition to the zero-shot setting, we investigate the effects of in-context learning for SummIt, as shown in the lower block of Table 2. The results consistently demonstrate that incorporating in-context learning significantly enhances the model's performance on ROUGE and G-Eval scores. This observation underscores the substantial few-shot capabilities of SummIt, showcasing its ability to adapt effectively and generate high-quality summaries in contexts with very few examples.

To further verify the summary quality, we conduct a human study to evaluate the overall quality of the summaries as shown in Table 3. According to the five-point **Likert scale ratings**, the summaries of ChatGPT and SummIt consistently outperform pre-trained language model results. The iterative refinement of SummIt also provides consistent improvements, which align with the G-Eval results obtained from the automatic evaluation. We also conducted a **human preference** study, where summaries from all models were presented to human annotators. They were tasked to select the best summary, without any prior knowledge of the origin of each summary. Consistent with the findings in (Goyal et al., 2022), the results reveal a clear preference among human annotators for summaries generated by large language models (LLMs) for both CNN (86%) and BBC (62%) style summaries. We also notice the summaries of ChatGPT after our iterative refinement (SummIt) show a significant improvement in human preference, with 18% and 14% percent improvements on CNN/DM and XSum datasets. The results demonstrate the effectiveness of refining generic summaries of our framework.

|  | R1 | R2 | RL | G-Eval | FactCC | DAE |
|---|---|---|---|---|---|---|
| ChatGPT | 21.61 | **5.98** | 17.60 | 3.47 | 28.00 | 10.34 |
| SummIt | 21.92 | 5.93 | 17.62 | 4.24 | 36.00 | 33.02 |
| ChatGPT-IE | **22.01** | 5.11 | **17.06** | 3.85 | **51.68** | **93.68** |
| SummIt-IE | 19.72 | 3.85 | 15.36 | **4.95** | 47.24 | 90.36 |

Table 4: Experimental results of incorporating knowledge extractor on summary quality and faithfulness on XSum dataset. -IE refers to the model integrated with OpenIE.

|  | R1 | R2 | RL | G-Eval | BM25 | DPR |
|---|---|---|---|---|---|---|
| ChatGPT | 30.01 | 8.94 | 27.03 | 1.06 | 33.09 | 77.22 |
| ChatGPT-Topic | **33.24** | **10.20** | **29.88** | 1.16 | 36.20 | 78.77 |
| SummIt-Topic | 30.45 | 8.48 | 27.19 | **4.74** | **39.11** | **82.41** |

Table 5: Experimental results on NEWTS dataset to test the controllability of our framework. -Topic indicates a model that is prompted to extract topic-related snippets before generating a summary.

### 4.3 Summary Faithfulness Evaluation

To evaluate the efficacy of the SummIt framework in enhancing summary faithfulness with the knowledge extractor, we conducted additional experiments, as presented in Table 4. The findings demonstrate that our framework's iterative approach to refining summaries yields significant improvements in summary faithfulness, as indicated by both FactCC and DAE results. Furthermore, the integration of a knowledge extractor such as OpenIE further enhances the level of faithfulness. The LLM-based evaluation score G-Eval also indicates a higher level of satisfaction with the refined summaries when guided by the extracted knowledge triplets. In conclusion, our study reveals that iterative refinements with the incorporation of the knowledge extractor effectively enhance summary faithfulness without compromising the quality of the summaries.

### 4.4 Query-focued Summarization Controlability Evaluation

We utilize the query-based summarization dataset NEWTS as our testbed to demonstrate the controllability ability of SummIt. The results obtained, as depicted in Table 5, highlight the framework's capability to align the focus of a generic summary with the specific topic of interest or query provided by the user. We also observe improved G-Eval evaluation scores by directing the summary generation process toward the intended topic.

Furthermore, we evaluate the controllability of the summarization systems by quantifying the sim-

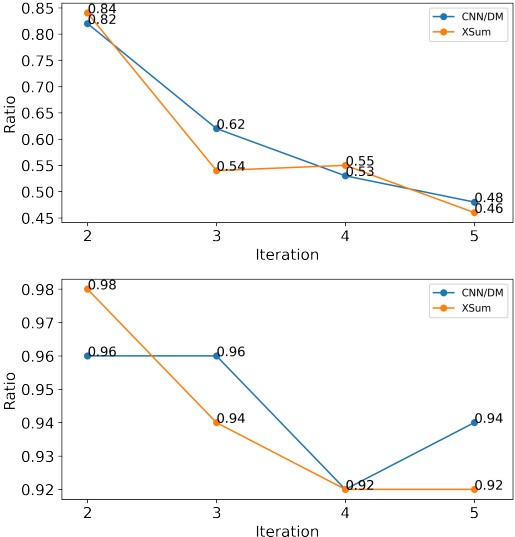

Figure 3: Human evaluation to justify the refinement behavior of SummIt. The top plot refers to the human justification of the ratio that the summary is improved at each iteration and the bottom plot indicates the ratio that the summarizer follows the evaluator's evaluation rationale.

ilarity between the query and the generated summary. Both BM25 and DPR are employed as similarity metrics, and we consistently observe enhancements after the iterative refinement process. This observation serves as evidence that SummIt effectively refines the summary to align with the topics specified in the query.

## 5 Analysis

### 5.1 Ablation Studies

Table 6 shows the results of the ablation study by removing refinement operations. The ablation study is conducted on the CNN/DM dataset under the zero-shot settings. According to the results, each option contributes to the success of our method, and the add operation affects the ROUGE score most, while the simplify operation affects the GPT-evaluation scores the most. Without adding the operation, the information in the iterative process will only decrease, resulting in less n-gram overlap. On the other hand, without the simplify and remove operations, the redundant information results in low G-Eval scores.

### 5.2 Over-correction Issue

A recent work (Liu et al., 2023) highlights the potential issue of LLM-based evaluators having a bias towards the LLM outputs, which raises the doubt:

|  | R1 | R2 | RL | G-Eval |
|---|---|---|---|---|
| SummIt | 36.50 | 13.49 | 26.76 | 4.33 |
| -w/o Add | 33.01 | 11.55 | 24.71 | 3.98 |
| -w/o Remove | 36.46 | 13.44 | 26.55 | 3.64 |
| -w/o Rephrase | 34.71 | 12.12 | 26.31 | 3.82 |
| -w/o Simplify | 33.49 | 12.33 | 25.76 | 3.55 |
| -w/o Keep | 33.87 | 13.03 | 25.70 | 3.94 |

Table 6: Ablation Study on Iterative Refinement Operations

**Summary-Iter1:** Hospitals in Wales may have to choose between emergency care and non-urgent surgery during peak winter months, according to Dr. Andrew Goodall. He suggested that hospitals may decide not to schedule surgery in order to focus on "front door pressures." Two hospitals in Swansea and Bridgend have already cancelled some surgical procedures until after Christmas.

**Evaluation-Iter1:** The summary effectively conveys the main point of the article, but it could be shortened for conciseness. Consider removing the specific hospitals mentioned and rephrasing the sentence about hospitals having to choose between emergency care and non-urgent surgery to make it more concise.

**Summary-Iter2:** Hospitals in Wales may have to prioritize emergency care over non-urgent surgery during peak winter months, according to Dr. Andrew Goodall. Some surgical procedures have already been cancelled until after Christmas.

**more iterations...**

Table 7: An example of iterative summary refinement from the XSum dataset. The revision between the two iterations and their corresponding comments are presented in the same color. The blue color refers to the rephrase revision and the orange color refers to the remove operation.

*1) Does the refinement actually improve the summary?*
Another potential issue of SummIt would be:

*2) Does the refinement actually follow the rationale feedback from the evaluator?*

To address these two concerns and provide further validation for the step-wise summary refinement in SummIt, we conducted the corresponding human evaluations. Specifically, we asked expert human annotators to label **1)** whether these edits resulted in improvements to the summary based on human judgment and **2)** whether the edits made by the summarizer align with the feedback provided in the last step by the evaluator.

The results of the human evaluation, presented in Figure 3, indicate that approximately 90% of the edits performed by the summarizer adhered to

the provided feedback as intended on both datasets. However, only around $50-60\%$ of these edits after 2 or more iterations were deemed beneficial according to human judgment, whereas the evaluator in SummIt still asks to perform the refinements. We also notice a clear trend that the percentage of beneficial refinements decreases as the iteration number goes up. The finding shows an **Over-correction problem:** the LLM may demand itself to continuously refine the summary based on its own evaluation criteria, rather than adhering to the true evaluation criteria of good summaries by humans.

This finding highlights the need for better stopping criteria in developing iterative summarization systems, and we argue that incorporating human-in-the-loop may be a potential solution. We leave this for future work.

### 5.3 Case Study

We show an example of iterative summary refinement in Table 7. The evaluator provides a detailed rationale for the summary and the summarizer can refine the summary accordingly. The full example can be found in Appendix C.

## 6 Conclusion

In this paper, we propose a new framework for text summarization by iteratively refining summaries with model feedback. Our framework is fully built upon large language models and doesn't require supervised training or reinforcement learning alignment. We also demonstrate that the system improves faithfulness and controllability by incorporating knowledge and topic extractors. We conduct extensive experiments and analyses on three benchmark datasets and experimental results show that our iterative summarization system outperforms the one-shot generation setting systems with LLM, which demonstrates the effectiveness of our method. Our human evaluation finds that the summary refinement by our framework can clearly follow the self-evaluation feedback, but is highly biased toward its own evaluation criteria, rather than human judgment. We believe the potential issue could be addressed with human-in-the-loop feedback. We hope the insights gained from this work can guide future research in building more powerful LLM-based summarization systems.

## Limitations

Instead of conducting experiments on the entire test set, we randomly sample 1000 examples from each dataset test set due to budget limits. Previous research efforts (Goyal et al., 2022; Zhang et al., 2023d) have also been limited in their testing of GPT-3 on a small number of instances.

We only use *gpt-3.5-turbo* model from openAI API as an instance of large language models. The focus of the paper is to explore the iterative summarization framework with LLM, but not compare different open and closed LLMs.

## Acknowledgement

This work is partially supported by NSF through grants IIS-1763365 and IIS-2106972.

We express our gratitude to the anonymous reviewers for their valuable reviews and feedback.

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

**Appendix**

## A   Experimental Setup

We use the official checkpoints of the baseline models BART, T5, and PEGASUS from Huggingface. We use *gpt-3.5-turbo* model[4] as the backbone LLM for both the generation and evaluation of summaries, keeping the temperature parameter at 0 to ensure reproducibility.

As for the datasets, we randomly sample 1000 samples with random seed 101 from the test set for both CNN/DM and XSum datasets and use the full test set for the NEWTS dataset. We also tune the LLM optimal prompt and hyperparameters on a dev set of 50 examples. Each discovery experiment was run three times, and the average result was used to mitigate the instability of small datasets.

## B   Prompts

Here we list prompts used in our experiments for extracted and generated summaries in Table 8 and Table 9. Note that according to OpenAI's document, the model could receive two categories of prompts: system prompt and user prompt, where the system prompt functions as the global instruction to initialize the model and the user prompt as the question proposed by users. In our experiment, we leverage both prompts to guide the model and select the best prompts on a dev set of 50 examples.

| Model | System Prompt |
|---|---|
| **Summarizer** | You are a summarizer that follows the output pattern. You revise the summary based on the given instructions. You follow all the instructions without commenting on them. Make sure the summary is concise and accurate. |
| **Evaluator** | You are a summary evaluator that follows the output pattern. You give scores for the summaries as well as revise suggestions. Your score should be corresponding to your suggestions. You suggestions can be: 
 1. Add the information of [] 
 2. Remove the information of [] 
 3. Rephrase the information of [] 
 4. Shorten the summary. 
 5. Do nothing. 
 Only ask for the information that appeared in the document. If you find the summary is too long, ask for a shorter summary. Keep the summary short and concise. If you think there's no further revision is needed, you must add "<STOP>" at the end of your output at the end of the comment. Give precise and clear suggestions. |

Table 8: System prompts of the summarizer and the evaluator for all settings.

## C   Example summaries

Here we show the full example of the iterative summarization and rationale feedback in Table 10 from CNN/DM dataset, together with their golden references.

---

[4]https://platform.openai.com/docs/guides/gpt/chat-completions-api

| Setting | Model | User Prompt |
|---|---|---|
| **Quality** | **Summarizer** | **Summarize**: [*In-context Examples*] Please summarize the following document. [*Document Content*] [*Format Instructions*] |
| | | **Refine**: [*Revise Suggestions*] Revise the summary. Follow all the suggestions and you can not make more comments. [*Format Instructions*] |
| | **Evaluator** | **Evaluate**: [*In-context Examples*]Please evaluate the summary for the document.[*Document Content*] [*Summary Content*].The output should be a probability distribution of assigning the score between 1-5 as well as its justification. Please give revise comments if you think this summary is not good enough.[*Format Instructions*] |
| **Control** | **Summarizer** | **Summarize**: [*In-context Examples*] Please summarize the following document based on the given topic sentence. [*Document Content*] [*Topic Sentence*] [*Format Instructions*] |
| | | **Refine**: [*Revise Suggestions*] Revise the summary. Follow all the suggestions and you can not make more comments. [*Format Instructions*] |
| | **Evaluator** | **Evaluate**: [*In-context Examples*] Please evaluate the summary for the document to check if the summary follows the given topic sentence.[*Document Content*] [*Summary Content*] [*Topic Sentence*].The output should be a probability distribution of assigning the score between 1-5 as well as its justification. Please give revise comments if you think this summary is not good enough.[*Format Instructions*] |
| **Faithfulness** | **Summarizer** | **Summarize**: [*In-context Examples*] Please summarize the following document based on the given relationships. [*Document Content*] [*OpenIE Relationships*] [*Format Instructions*] |
| | | **Refine**: [*Revise Suggestions*] Revise the summary. Follow all the suggestions and you can not make more comments. [*Format Instructions*] |
| | **Evaluator** | **Evaluate**: [*In-context Examples*] Please evaluate the summary for the document to check if the summary follows the given relationships.[*Document Content*] [*Summary Content*] [*OpenIE Relationships*].The output should be a probability distribution of assigning the score between 1-5 as well as its justification. Please give revise comments if you think this summary is not good enough.[*Format Instructions*] |

Table 9: User prompts of summarizer and evaluator for different settings.

| Document | Media playback is unsupported on your device16 december 2014 last updated at 08:58 gmtdr andrew goodall said a choice may have to be made between emergency care and non-urgent surgery .in an extended interview with bbc wales health correspondent owain clarke , dr goodall said hospitals might make the decision not to schedule surgery in the first place to focus on " front door pressures " during the peak winter months .two hospitals , morriston in swansea and the princess of wales in bridgend , have already cancelled some surgical procedures until after christmas . |
|---|---|
| Reference | The new chief executive of the welsh nhs has told bbc wales winter pressures could force hospitals to cancel operations. |
| Summary-Iter1 | Hospitals in Wales may have to choose between emergency care and non-urgent surgery during peak winter months, according to Dr. Andrew Goodall. He suggested that hospitals may decide not to schedule surgery in order to focus on front door pressures.Two hospitals in Swansea and Bridgend have already cancelled some surgical procedures until after Christmas. |
| Evaluation-Iter1 | The summary effectively conveys the main point of the article, but it could be shortened for conciseness. Consider removing the specific hospitals mentioned and rephrasing the sentence about hospitals having to choose between emergency care and non-urgent surgery to make it more concise. |
| Summary-Iter2 | Hospitals in Wales may have to prioritize emergency care over non-urgent surgery during peak winter months, according to Dr. Andrew Goodall. Some surgical procedures have already been cancelled until after Christmas. |
| Evaluation-Iter2 | The revised summary effectively conveys the main point of the article in a concise manner. However, it could benefit from adding more specific information about the reason for the cancellations and the potential impact on patients. Consider adding information about the reason for the cancellations and the potential impact on patients to make the summary more informative. |
| Summary-Iter3 | Dr. Andrew Goodall warns that hospitals in Wales may have to prioritize emergency care over non-urgent surgery during peak winter months, which has already led to some surgical procedures being cancelled until after Christmas. This decision is made to focus on front door pressuresänd ensure that emergency care is not compromised. The cancellations may cause inconvenience to patients, but it is necessary to ensure that emergency care is not overwhelmed. |
| Evaluation-Iter3 | The revised summary effectively conveys the main point of the article in a concise and informative manner. It includes specific information about the reason for the cancellations and the potential impact on patients, as well as the importance of prioritizing emergency care. The only suggestion would be to consider shortening the summary slightly for even greater conciseness. |
| Summary-Iter4 | Hospitals in Wales may have to prioritize emergency care over non-urgent surgery during peak winter months, leading to some cancellations until after Christmas. Dr. Andrew Goodall emphasizes the importance of focusing on front door pressuresïo ensure that emergency care is not compromised. While the cancellations may cause inconvenience to patients, it is necessary to prevent emergency care from being overwhelmed. |
| Evaluation-Iter4 | The revised summary effectively conveys the main point of the article in a concise and informative manner. It includes specific information about the reason for the cancellations and the potential impact on patients, as well as the importance of prioritizing emergency care. The summary is well-written and does not require any further revision. |

Table 10: Case study