# OpenReview forum: "SummIt: Iterative Text Summarization via ChatGPT"
_EMNLP/2023/Conference — EMNLP 2023 Findings_

### Official Review · Reviewer_h2TX · 2023-08-03

**Soundness:** 3

**Excitement:**

4: Strong: This paper deepens the understanding of some phenomenon or lowers the barriers to an existing research direction.

**Paper Topic And Main Contributions:**

The paper proposes SummIt, an LLM (specifically, ChatGPT) based summarization framework with iterative summary generation as opposed to single-pass adopted in previous work. For the subsequent pass, the authors employ LLM self-evaluation and feedback for the model to improve the output.
The feedback is limited to certain edit actions, and the key document content to edit is provided by a separate knowledge extraction model.

The authors compare SummIt to a series of LM and LLM-based approaches working in a traditional 1-pass setting in terms of human evaluation along several qualitative dimensions. On CNN/DM, in SummIt elicits the best performance on most metrics (including Overall); on XSum, ChatGPT is more fluent, but Overall, SummIt outperforms it as well. Human preference is given to SummIt on both datasets.

In further evaluation of the factuality of the model equipped with the knowledge extractor (G-Eval, FactCC, DAE metrics), ChatGPT without iterative refinements performed better in terms of FactCC and DAE, SummIt being better on G-Eval. In the controllability evaluation of query-focused summarization on NEWTS dataset, SummIt-Topic (the model with additional topic-related content in the prompt), outperforms 1-pass ChatGPT on G-Eval, BM25, and DPR metrics.

Finally, a separate human evaluation shows that the summaries improve with subsequent iterations, and the summarizer follows the evaluator's rationale.

**Reasons To Accept:**

* a simple and efficient technique SummIt for iterative summarization improvement with an LLM is presented
* SummIt attains strong humans' preference in the corresponding evaluation round, performs competitively on automatic metrics

**Reasons To Reject:**

* knowledge extractor part shows mixed results in the faithfulness automatic evaluation

**Reproducibility:**

4: Could mostly reproduce the results, but there may be some variation because of sample variance or minor variations in their interpretation of the protocol or method.

**Reviewer Confidence:**

4: Quite sure. I tried to check the important points carefully. It's unlikely, though conceivable, that I missed something that should affect my ratings.

---

> ### Author Rebuttal · Authors · 2023-08-28
>
> We sincerely value the priceless evaluation of our paper and the devoted time you've dedicated to offering valuable feedback. We have thoroughly assessed each of your points and are eager to provide thorough responses that address them comprehensively:
>
> **Question: knowledge extractor part shows mixed results in the faithfulness automatic evaluation**
>
> **Answer** We agree with this point. Our stance is that there might exist certain **trade-offs** between the faithfulness of the summary and the overall quality of the summary. During the iterative refinements, the LM may sacrifice some faithfulness to improve the overall summary quality. However, compared to the case without a knowledge extractor, we did notice substantial faithfulness performance gaps, indicating the advantages of guiding summary generation with extracted knowledge. Overall, SummIt with the knowledge extractors could generate high-quality and faithful summaries.
>
> **Question: important details of human evaluations left unknown: were there multiple passes, what is inter-annotator agreement on those**
>
> **Answer:** Thanks for pointing this out. We will definitely add more human evaluation details in the appendix in our final version. We conducted the human evaluation in a single pass with multiple annotators, and below is a table summarizing the inter-annotator agreement in terms of different evaluation aspects during our annotation process. We used **Pearson Correlation** to evaluate the inter-annotator agreement
>
> **Inter-annotator agreement**
> | Aspect         |  correlation    |
> | --------------- | ----- |
> | **Coherence**   | 0.658 |
> | **Fluency**     | 0.851 |
> | **Relevance**   | 0.796 |
> | **Consistency** | 0.621 |
> | **Conciseness** | 0.747 |
> | **Overall**     | 0.742 |
>
> We will release our annotations to better improve reproducibility.
>
>
> **Question: Given ChatGPT's inherent stochasticity, it would be essential to have public access to the summaries for reproducibility, which I didn't see offered in this work**
>
> **Answer:** We totally agree with the point. We are committed to reproducibility and will definitely open-source all the source codes, generation outputs, and human studies upon acceptance. We included our detailed experimental Setup, prompts, and example outputs in the appendix.

---

### Official Review · Reviewer_Ceeh · 2023-08-05

**Typos Grammar Style And Presentation Improvements:** 1. G-Eval, FactCC, and DAE are not ve…
**Soundness:** 3

**Excitement:**

4: Strong: This paper deepens the understanding of some phenomenon or lowers the barriers to an existing research direction.

**Paper Topic And Main Contributions:**

The paper proposes a new model SummIt which iteratively refines its summarization by the prompt generated by its evaluator. And the paper shows that knowledge and topic extractors improve faithfulness and controllability. SummIt exceeds SOTA models in some metrics.

**Questions For The Authors:**

A. For the L235, have you considered using y^i as input since "Rephrase: Rephrase the information of <insert> in the summary" is rephrasing the corresponding content in the summary?

**Reasons To Accept:**

1. The paper is well-written and thorough experiments are made with several meaningful metrics, not just ROUGE score. The proposed model exceeds SOTA models in several aspects.
2. The direction is promising and can be used in text summarization.

**Reasons To Reject:**

1. The model has a lower ROUGE score compared to other models. And the model only compares to the model with zero-shot and few-shot learning. The performance is much lower compared to the fine-tuning model on the tasks. I am wondering what's the upper bound performance of the model if it can be fine-tuned.
2. The experiments are done on limited data points. Not sure if the method can adapt to more complex scenarios other than CNN and XSum.

**Reproducibility:**

4: Could mostly reproduce the results, but there may be some variation because of sample variance or minor variations in their interpretation of the protocol or method.

**Reviewer Confidence:**

2: Willing to defend my evaluation, but it is fairly likely that I missed some details, didn't understand some central points, or can't be sure about the novelty of the work.

---

> ### Author Rebuttal · Authors · 2023-08-28
>
> We greatly appreciate the meticulous review of our paper and the dedicated effort you've invested in providing invaluable feedback. We have conscientiously examined each of your concerns and would like to offer comprehensive responses to address them:
>
> **Question: The model has a lower ROUGE score compared to other models. And the model only compares to the model with zero-shot and few-shot learning. The performance is much lower compared to the fine-tuning model on the tasks. I am wondering what's the upper bound performance of the model if it can be fine-tuned.**
>
> **Answer:** We agree the zero-shot and few-shot results are much lower compared to the fine-tuned models like PEGASUS and BART. This discrepancy is probably attributed to the fact that training with similar data during fine-tuning may hint the model about the style, length, and coverage of the summaries. Unfortunately, our backbone is ChatGPT, which is not open-sourced, so we can't directly fine-tune the model. We will explore our method in fine-tuning open LLMs like LLaMa in the future.
>
> **Question: The experiments are done on limited data points. Not sure if the method can adapt to more complex scenarios other than CNN and XSum.**
>
> **Answer:** Thanks for pointing this out. CNN and XSum are both news domain summarization benchmarks and we agree testing our framework on other domains and longer documents would better validate our approach.
>
> Below is the Pubmed dataset (209.5 words/summary) results of ChatGPT and our Summit framework.
>
> **PubMed Experimental results**
>
> | Setting   | R1    | R2    | RL    | G-Eval |
> | --------- | ----- | ----- | ----- | ------ |
> | ChatGPT   | 31.92 | 12.01 | 26.44 | 3.18   |
> | Summit(Zero-shot) | 26.56 | 10.32 | 21.07 | 3.39   |
> | Summit(Few-shot) | 27.33 | 10.01 | 22.56 | 3.58   |
>
> We notice similar trends as in the news domain, that our Summit consistently improves the GPT-evaluation scores while showing lower ROUGE scores compared to ChatGPT. On the other hand, few-shot examples benefit the model in terms of both metrics on long summarization generation on the PubMed dataset.
>
> **Question: For the L235, have you considered using y^i as input since "Rephrase: Rephrase the information of <insert> in the summary" is rephrasing the corresponding content in the summary?**
>
> **Answer:** Thanks for bringing up this interesting idea. We tried this option in our experiment and empirically found the performance is worse than our current setting. Using y^i in generating the next round y^{i+1} will confuse ChatGPT and make the output more similar. We will include this analysis in our final version.
>
> **Question: G-Eval, FactCC, and DAE are not very common metrics for summarization. Can you give some explanations in the appendix?**
>
> **Answer:** Thanks for pointing this out. We briefly introduce these metrics in the Evaluation metrics in Section 4.1 due to space limit. We will add more explanations and details in the appendix in our final version.
>
> Specifically, G-Eval is a very recent summary quality evaluation matrix based on large language models. G-Eval uses LLM with chain-of-thoughts (CoT) and a form-filling paradigm to assess the quality of NLG outputs. It shows the highest correlation with humans compared to other summarization quality metrics.
>
> FactCC is a weakly-supervised BERT-based model metric that verifies factual consistency through rule-based transformations applied to source document sentences. It shows a high correlation in assessing summary faithfulness with human judgments.
>
> DAE (Defining arc entailment) decomposes entailment at the level of dependency arcs, examining the semantic relationships within the generated output and input. Rather than focusing on aggregate decisions, DAE measures the
> semantic relationship manifested by individual dependency arcs in the generated output is
> supported by the input.

---

### Official Review · Reviewer_dwUH · 2023-08-11

**Soundness:** 3

**Excitement:**

3: Ambivalent: It has merits (e.g., it reports state-of-the-art results, the idea is nice), but there are key weaknesses (e.g., it describes incremental work), and it can significantly benefit from another round of revision. However, I won't object to accepting it if my co-reviewers champion it.

**Paper Topic And Main Contributions:**

They introduce "SummIt," an iterative text summarization framework leveraging large language models, such as ChatGPT. Their approach allows the model to continuously refine its summaries by employing self-assessment and feedback, mirroring the human method of drafting and revising summaries. Additionally, they investigate the advantages of incorporating knowledge and topic extractors into their framework.

**Reasons To Accept:**

1. The paper is well-written and easy to follow.
2. They conduct experiments on three summarization benchmark datasets. Although their proposed approach is not complicated, it exceeds SOTA models in some aspects.
3. This approach can be further developed and improved for text summarizations.

**Reasons To Reject:**

you used 5 refinement operations, but we don't know the effects of each operation. More analysis for this are necessary. For example, if removing "Keep the summary unchanged", would the results be different?

**Reproducibility:**

4: Could mostly reproduce the results, but there may be some variation because of sample variance or minor variations in their interpretation of the protocol or method.

**Reviewer Confidence:**

3: Pretty sure, but there's a chance I missed something. Although I have a good feel for this area in general, I did not carefully check the paper's details, e.g., the math, experimental design, or novelty.

---

> ### Author Rebuttal · Authors · 2023-08-28
>
> Thanks for your thorough review of our paper and the effort you've put into providing valuable feedback. We have carefully considered your concern, and we would like to address comprehensively:
>
> **Question: You used 5 refinement operations, but we don't know the effects of each operation. More analysis for this is necessary. For example, if removing "Keep the summary unchanged", would the results be different?**
>
> **Answer:** Thank you for bringing up this point. We agree that including an ablation study involving removing options would more effectively showcase the performance of our system. The table below shows the ablation study results. We will also include it in the final version of the paper.
>
>
> **Table Refinement Operations Ablation Analysis**
>
> |              | R1    | R2    | RL    | G-Eval |
> | ------------ | ----- | ----- | ----- | ------ |
> | SummIt       | 36.50 | 13.49 | 26.76 | 4.33   |
> | w/o Add      | 33.01 | 11.55 | 24.71 | 3.98   |
> | w/o Remove   | 36.46 | 13.44 | 26.55 | 3.64   |
> | w/o Rephrase | 34.71 | 12.12 | 26.31 | 3.82   |
> | w/o Simplify | 33.49 | 12.33 | 25.76 | 3.55   |
> | w/o Keep     | 33.87 | 13.03 | 25.70 | 3.94   |
>
> The ablation study is conducted on the CNN/DM dataset under the zero-shot settings. According to the results, each option contributes to the success of our method, and the add operation affects the ROUGE score most, while the simplify operation affects the GPT-evaluation scores the most. Without adding the operation, the information in the iterative process will only decrease, resulting in less n-gram overlap. On the other hand, without the simplify and remove operations, redundant information, resulting in low G-Eval scores.
>
> We hope the results could address your concern about our paper. Thanks again for the helpful feedback.

---

### Meta-Review · Area_Chair_etrn · 2023-09-19

**Recommendation:** 2

**Metareview:**

The paper introduces SummIt, which iteratively refines its outputs using LLM self-evaluation and feedback. The paper is well-written with thorough experiments, the approach performs competitively on automatic metrics and also receives positive feedback during human evaluation. While the approach shows promise, reviewers highlighted some concerns about the model's adaptability beyond CNN and XSum, its comparison to the fine-tuning approach, and mixed results on the faithfulness evaluation.

---

### Decision · Program_Chairs · 2023-10-07

**Decision:**

Accept-Findings

**Comment:**

The paper introduces SummIt, which iteratively refines its outputs using LLM self-evaluation and feedback. The paper is well-written with thorough experiments, the approach performs competitively on automatic metrics and also receives positive feedback during human evaluation. While the approach shows promise, reviewers highlighted some concerns about the model's adaptability beyond CNN and XSum, its comparison to the fine-tuning approach, and mixed results on the faithfulness evaluation.